# Severity of Food Insecurity among Australian University Students, Professional and Academic Staff

**DOI:** 10.3390/nu14193956

**Published:** 2022-09-23

**Authors:** Katherine Kent, Denis Visentin, Corey Peterson, Ian Ayre, Catherine Elliott, Carmen Primo, Sandra Murray

**Affiliations:** 1School of Health Sciences, Western Sydney University, Penrith, NSW 2751, Australia; 2School of Health Sciences, University of Tasmania, Launceston, TAS 7250, Australia; 3Sustainability Unit, University of Tasmania, Launceston, TAS 7250, Australia

**Keywords:** food insecurity, food security, university students, college students, university staff

## Abstract

Assessments of the severity of food insecurity within Australian university students are lacking, and the experience of food insecurity in Australian university staff is unknown. A cross-sectional online survey in March 2022 aimed to characterize the severity of food insecurity in students, professional and academic staff at the University of Tasmania (UTAS). The Household Food Security Survey Module six-item short form assessed food security status in addition to seven demographic and education characteristics for students and six demographic and employment characteristics for staff. Participants were categorized as having high, marginal, low, or very low food security. Multivariate binary logistic regression identified students and staff at higher risk of food insecurity. Among student respondents (*n* = 1257), the prevalence of food insecurity was 41.9% comprising 8.2% marginal, 16.5% low, and 17.3% very low food security. Younger, non-binary, first-year enrolled, on campus, and international students were at significantly higher risk of food insecurity. Among staff (*n* = 560), 16.3% were food insecure comprising 3.8% marginal, 5.5% low, and 7.0% very low food security. Professional staff, staff on casual contracts, and staff recently employed, were at significantly higher risk of food insecurity. Our findings suggest a high occurrence of food insecurity in UTAS students and staff, with a large proportion of food insecure staff and students experiencing very low food security. Our findings have implications for efforts towards reducing food insecurity at university campuses through a holistic and integrated approach, advocating for food systems that support healthy, sustainable, and equitable food procurement and provision for both university students and staff.

## 1. Introduction

Food security is said to exist when all people, at all times, have physical, social and economic access to sufficient safe and nutritious food that meets their dietary needs and food preferences for an active and healthy life [1,2] In contrast, food insecurity occurs if these needs are not met or if food is not accessible in a socially acceptable way. Food insecurity is considered to range in severity from experiencing anxiety that food will run out, to reduction of the quality, variety and amount of food consumed, or regularly going without food at all [3]. Food security is multi-dimensional, encompassing six pillars of (1) food availability; (2) food access (including financial and physical access), (3) food utilisation (processing and consumption of food) (4) stability (stability in the other pillars over time) (5) agency (the capacity to make decisions about food), and (6) sustainability (viability of the ecological and social bases of food systems) [2]. In Australia, the prevalence of food insecurity varies according to the population being studied, and has conservatively been estimated to be 5% across the entire Australian population [4]. Despite ongoing research documenting the growing prevalence and negative experiences of food insecurity, there has been insufficient progress related to United Nations (UN) Sustainable Development Goal (SDG) 2 ‘Zero Hunger’, and the world is not on track to achieve this goal by 2030, even in high income countries like Australia [5]. Food security is impossible without a shift to a more healthy, sustainable and equitable food system.

Transitioning towards sustainable food systems for the population and planet will require governments and institutions to develop effective governance to support the adoption of sustainable food practices [6]. Universities have made progress in the development and implementation of sustainability plans and programs related to energy use, housing, and transportation to address issues of environmental and economic sustainability [6]. However, these plans may inadvertently omit social sustainability, such as the effects of university students and staff food insecurity. The University of Tasmania (UTAS) Sustainability Survey is a biennial survey developed to gain insight into student perceptions, aspirations and behaviours on sustainability and inform the development and implementation of sustainability initiatives and programs. In recognition of social sustainability efforts, recent UTAS Sustainability surveys have included questions on food security.

University students across the world have been shown to be at higher risk of food insecurity than the general population, at between 35% and 42% of students [7]. While research has predominantly focussed on undergraduate populations, research has also identified a high prevalence of food insecurity in postgraduate students [8]. University students who are food insecure have poorer physical and mental health outcomes, eat poorer diets [9], and experience worse academic performance [9]. The culmination of these factors may reduce a students’ ability to continue their university studies [10]. In early March 2020, the UTAS Student Sustainability Survey determined food insecurity in a regional Australian university student population, identifying that 38% of all UTAS students were food insecure [11]. However, this preliminary study was limited through its utilisation of a single-item measure of food insecurity, which may underestimate the prevalence of food insecurity, and was limited in its ability to determine the severity of food insecurity experienced by students [12]. Further research using a more comprehensive tool to determine food insecurity, including a tool that can determine the severity of food insecurity is warranted. Additionally, the potential experience of food insecurity in the wider university community such as staff was not considered in the 2020 UTAS survey which should be a consideration for future research.

Across the world, the occurrence and impact of food insecurity in the wider university community, including staff, has been less well studied. Two studies in international literature have documented that the prevalence of food insecurity in university faculty (academic staff) in Vermont is between 2–5% depending on the timing of the academic year, and in other university staff (such as professional staff) the prevalence of food insecurity is between 11–14% [13]. This number is dwarfed by the estimated 70% of professional staff who were reported to experience food insecurity at another American college [14]. A clearer understanding of the experience of food insecurity across a whole university campus in an Australian setting is yet to be documented despite the role that university campuses and on-campus food outlets could play in supporting food security in these groups.

Immediately after the UTAS Student Sustainability Survey in 2020, the COVID-19 pandemic impacted food access and supply in Australia [15,16], in addition to having severe economic impacts for businesses and households [17]. Many Australians experienced food insecurity for the first time [18], and households had additional challenges putting healthy food on the table [19]. Unsurprisingly, the prevalence of food insecurity in student populations worsened both in Australia and internationally [20,21,22] which was related to losing employment, increase of food prices, and a perceived threat posed by COVID-19 in shopping for food [23]. In Australia, many students lost their temporary jobs, often in hospitality or retail settings that were closed during lockdowns [24]. In addition, the pandemic triggered widespread disruption in the lives of university students, such as shifting to online learning with impacts on mental health and wellbeing [25]. Further, Australian universities lost a substantial amount of revenue (between AU$1.82 billion to AU$36 billion) [26] driven mainly by up to 25% lower enrolments from international students, while Australian universities and their staff were also ineligible for government social support mechanisms (such as the JobKeeper program paid to employers to retain staff) [27]. Consequently, universities experienced a loss of between 10% and 20% of the FTE workforce from 2019–20 [28]. Although variable across universities, the impact on staffing levels disproportionately impacted workers employed on casual and fixed-term contracts. For example, the percentage decrease in casual staff was especially dire, with the percentage of casual workers decreasing between 19–59% [28]. In 2022, Australian borders reopened for international travellers including students, lockdowns ceased, and most universities returned to varying levels of in-person classes [27] although not always at pre-pandemic levels. Following this return to a ‘new COVID normal’, within universities, it is important to re-examine the issue of food insecurity within the higher education setting, to ensure the entire university community has access to enough safe and healthy food.

As our previous research could not determine the severity of food insecurity and excluded staff from the analyses, coupled with the potential for world events to increase food insecurity in Australian university students and staff, further investigation into the severity and demographic predictors of food insecurity in both university student and staff populations is required. In particular, by identifying the particular demographics of student and staff populations at higher risk of food insecurity, interventions to mitigate food insecurity risks in these specific populations could be prioritised. Therefore, it was the aim of this study to characterise the severity of food insecurity among students, professional staff, and academic staff at a regional Australian university, and to identify the demographic groups of students and staff at higher risk of food insecurity.

## 2. Materials and Methods

### 2.1. Study Setting and Participants

UTAS is a public research university with three main Tasmanian campuses (Hobart, Launceston, and Burnie), and one campus in Rozelle in New South Wales, Australia. The 2022 UTAS Student and Staff Sustainability Surveys were the fourth biennial sustainability surveys at UTAS. The staff survey is a proprietary survey called the Sustainability Culture Indicator (SCI) from Awake P.L., but also allows for UTAS-specific questions (e.g., food security). The student survey uses most of the same questions under an agreement with the SCI authors, but with more additional university-specific questions (also including food security). The primary purpose of the Student and Staff Sustainability Surveys is to track perceptions of staff and students regarding the University’s increasing commitment to the UN SDGs, planetary health and human flourishing [29] as well as provide feedback on areas in which the University should focus its efforts to achieve more holistic sustainability outcomes.

### 2.2. Questionnaire Development

The U.S. Household Food Security Survey Module Six-Item Short Form (HFSSM) was used to assess food security status in both the student and staff surveys [30]. This validated screening tool [31] seeks responses to six questions that self-report uncertain, insufficient or inadequate food access, availability and utilization due to limited financial resources, and the compromised food consumption that may result. Participant responses to the six questions were coded and assessed in accordance with the user notes [32], where each affirmative response was assigned a score of 1, and summed raw scores were used to describe the severity of food insecurity. Scores were then used to categorize respondents as having high (0), marginal (1), low (2–4) or very low food security (5–6). The USDA HFSSM user notes suggest that food insecurity is classified as a score of 2 or more. However, this study combines marginal food security (a single instance of food insecurity) in our classification of ‘food insecure’ which is in line with recommendations from some research teams internationally to classify marginal food security as food insecure [33], including the latest national food security reports in Canada [34]. This recommendation is related to the increased anxiety over food procurement and lack of agency over food decisions that underpins marginal food security. Indeed, recent estimates of the prevalence of food insecurity in Tasmania have adopted this approach [18,19]. Further, published research in student populations has recommended that students experiencing marginal food security should not be grouped with students experiencing high food security due to poor outcomes in this group [35]. Therefore, a binary variable was also generated for food secure respondents (a score of 0 or high food security) or food insecure respondents (score of 1 or more comprising marginal, low and very low food security groups). Seven demographic and education characteristics of students were collected; age category, gender (male, female, non-binary, self-identify), length of study (first year to third year or more), degree type (pre-degree/short course, undergraduate or postgraduate), the region of the campus they usually attend (South, North, North-West, Sydney) primary mode of study (online or on-campus) and enrollment type (domestic or international student). Six demographic and employment characteristics of staff collected included age category, gender, role type (professional services, academic, professional and academic), employment contract type (tenured/permanent, fixed contract, casual, adjunct/honorary), the region of the campus of employment, and length of employment (<1 year to 10+ years).

### 2.3. Data Collection

All UTAS students enrolled at the time of the survey (*n* = 31,570) and all staff (*n* = 2946) were invited to participate via email to complete the online survey. Data collection occurred from 7–20 March 2022. Recruitment involved the promotion of the surveys through internal emails and social media channels. The survey was hosted using the online survey platform Qualtrics for the student survey and an out-sourced platform used by Awake P.L. (QuestionPro) as the staff survey provider. All participants were provided with a participant information sheet on the first page of the survey. Participants’ consent was implied by proceeding from the information page to the survey proper (as explained to them in the information page).

### 2.4. Data Analysis

Data were exported from online survey platforms and prepared for statistical analysis. All available survey data was used in the analyses. Data were analyzed using IBM SPSS Statistics for Windows, Version 27.0 (IBM Corp., Armonk, NY, USA).

For both staff and student surveys, all demographic, education, and employment variables were either categorical or ordinal and reported as frequencies and proportions. Cross-tabulations were employed to generate descriptive statistics related to food security status and with each of the demographic, education, and employment variables. Univariate logistic regression was performed individually for each demographic, education, and employment variable to generate unadjusted odds ratios for the binary food insecurity variable. Multivariable logistic regression was performed including all variables that were associated with food insecurity in the univariate analyses. The level *p* < 0.1 was used to include and retain variables in the models to yield adjusted odds ratios for food insecurity. The significance level for all analyses was set at *p* ≤ 0.05.

## 3. Results

### 3.1. Prevalence and Demographic Correlates of Food Insecurity in University Students

In total the student survey received 1257 responses, which is approximately 3.9% of the UTAS student cohort. Key demographic and education characteristics of the survey respondents, according to food security status are presented in Table 1. Among survey participants, students were predominantly aged between 18–24 years (43%), with a total of 69% self-reporting as female. Nearly half (42%) were enrolled in their first year of study and 45% were in undergraduate courses. The majority of students were studying as ‘on-campus students’ (57%) and most (85%) were domestic students. According to the six-item HFSSM, 41.9% (*n* = 527) of student respondents were classified as experiencing food insecurity (Figure 1). Of these, 8.2% of students were marginally food secure, 16.5% had low food security, while a further 17.3% of students experienced very low food security. For the individual items in the six-item HFSSM, 22.3% of participants reported that “The food that I bought just didn’t last and I didn’t have enough money to get more”, while 31.4% reported that “I couldn’t afford to eat balanced meals”. Nearly a third of participants (30.7%) had cut the size of their meals or skipped meals because there was not enough money for food. Almost one third of participants (31.4%) affirmed the statement “In the last 12 months did you ever eat less than you felt you should because there wasn’t enough money for food?”, and nearly a quarter of participants reported yes to the statement “In the last 12 months were you ever go hungry but didn’t eat because there wasn’t enough money for food?” (23.0%).

Table 2 presents crude and adjusted odds ratios of student food insecurity for the variables considered. The multivariate model retained the variables age, gender, years of enrolment, mode of study (distance/online) and enrolment status (international/domestic). The adjusted model had Pseudo R^2^ = 0.055, and Likelihood ratio test statistics χ^2^ = 1628.9, *p* < 0.001. The prevalence of food insecurity was 48% for students aged 18–24, and 47% for students aged 25–34 (Table 1). In comparison with students aged 55 years and over, younger students aged 18–24 years were 2.2 times more likely to be food insecure and students aged 25–34 years were 2.1 times more likely to be food insecure, adjusting for other factors (Table 2). Most students (69%) who did not identify as either male or female reported being food insecure (Table 1). In comparison to male and female identifying students, these students were nearly 3.5 times more likely to be food insecure in the multivariate model (Table 2). Most international students (61%) were classified as food insecure compared with 34% of domestic students (Table 1). International students were twice as likely to be food insecure compared to domestic enrolled students in the multivariate regression model (Table 2). Nearly half (48%) of on campus students were food insecure compared to a third (33%) of distance enrolled students. This was approaching significance in the multivariate analysis and may have been correlated to being an international student. Nearly half (45%) of first year students were classified as food insecure compared with 41% of second year and 38% of third year students (Table 1). First year students were 40% more likely to be food insecure compared to students who have been studying at UTAS 3 years or more when adjusting for other factors (Table 2).

### 3.2. Prevalence and Demographic Correlates of Food Insecurity in University Staff

In total, 560 UTAS staff completed the six-item HFSSM, which is approximately 19.0% of the UTAS staff cohort. As identified in Table 3, staff respondents to the survey were predominantly aged between 35–44 years and 45–54 years (both 28%). Most identified as female (55%), worked in professional staff roles (59%) and had a tenured or permanent employment contract (64%). More than a third had been employed at UTAS for a decade or more (34%).

Of all respondents, 16.3% (*n* = 91) reported experiencing some degree of food insecurity (Figure 1). Of these, 3.8% experienced marginal food security, 5.5% experienced low food security and 7.0% experienced very low food security (Figure 1). For the individual items in the 6-item HFSSM, 9.3% of participants reported an affirmative response to the statement “The food that I bought just didn’t last and I didn’t have enough money to get more”, while 11.6% reported an affirmative response to “I couldn’t afford to eat balanced meals”. 10.2% of staff reported that within the last 12 months they had cut the size of their meals or skipped meals because there was not enough money for food. 12.1% of staff reported yes to the statements “In the last 12 months did you ever eat less than you felt you should because there wasn’t enough money for food?”, and 9.6% reported yes to “In the last 12 months were you ever go hungry but didn’t eat because there wasn’t enough money for food?”.

Table 4 presents crude and adjusted odds ratios of staff food insecurity for the demographic and employment variables considered. In the multivariate analysis, the variables type of role, employment contract and length of employment were retained. The adjusted model had Pseudo R^2^ = 0.053, and Likelihood ratio test statistics χ^2^ = 468.9 *p* < 0.0001. In relation to role type, 11% of academic staff and 19% of professional staff experienced food insecurity (Table 3). Professional staff were 80% more likely to experience food insecurity compared to academic staff members in the multivariate model (Table 4). In terms of employment contract type, 35% of casual staff experienced some degree of food insecurity (Table 3) and were nearly 2.5 times more likely to experience food insecurity compared to tenured or permanent staff after adjusting for other factors (Table 4). Food insecurity differed by length of employment for UTAS staff members, where 31% of staff who had been employed for less than a year and 25% of staff who were employed for one to three years’ experienced food insecurity (Table 3). In comparison to those who had been employed for ten years or more, staff employed for less than a year were 3.6 times more likely to be food insecure and staff employed for between one to three years were 2.8 times more likely to be food insecure in the multivariate model.

Of professional staff, 17% of permanent staff were food insecure compared with 23% of fixed term contract and 26% of casual contract staff (Figure 2). A larger difference in the prevalence of food insecurity was evident for academic staff, where only 6% of tenured staff and 13% of academic staff on fixed-term contracts experienced food insecurity compared with 41% of academic staff on casual contracts.

## 4. Discussion

This study is the first, to our knowledge, to determine the severity of food insecurity in a sample of both university students, professional staff, and academic staff in an Australian university. Our study found an overall occurrence of food insecurity of 42% for students, which joins the statistics in a growing body of literature that shows that food insecurity is more prevalent for university students when compared to the general population [8,11,13,36]. Our study highlights that some groups of students, such as first year, on-campus, international students and non-binary identifying students are at higher risk of food insecurity, which have been overlooked in some previous Australian research. Our study also provides a novel contribution by providing the first statistics of food insecurity in university staff in Australia (16% of our sample of staff) and identifying that professional staff and staff on casual contracts (especially academic staff on casual contracts) are at substantially higher risk of food insecurity.

The occurrence of food insecurity in our study of students is slightly higher than the statistic of 38% captured by the 2020 UTAS Student Sustainability Survey [11]. Our results extend this previous study by showing that most food insecure students experience low and very low food security, indicating that they are regularly running out of food, eating poorer quality food, and going hungry. Our results also demonstrate that the measures put into place to support students during the COVID-19 pandemic, such as community days with free meals, emergency funding relief, and linking students with emergency food relief agencies through flyers have not fully addressed the experience of student food insecurity systemically or in the longer term. The prevalence of food insecurity for both students and staff was similar across the different regions of Tasmania, which indicates that interventions to support student and staff food security must be prioritised across all campuses.

Our study shows that first year students are at higher risk of food insecurity, contrasting some literature in US settings where it has been reported that food insecurity significantly increases when students enter their third or fourth year of university due to a transition to off campus accommodation [13]. This, coupled with our finding that students aged 35 years or younger are at the highest risk of food insecurity, identifies that in the Australian context students might be uniquely susceptible to food insecurity throughout their period of transition from school to university, which in Australia can coincide with a transition from the family home into living independently [36,37]. Our findings are aligned with previous Australian research that suggests commencing university and moving away from parents may be key times for intervention [8]. Unlike previous research, we did not demonstrate a difference in food insecurity between undergraduate and post-graduate university students [8]. In our study, international students were also at nearly twice the risk of food insecurity compared to domestic students. In Australia, international students have previously shown to be at substantially greater risk for food insecurity due to poor food literacy (such as cooking and grocery shopping skills), limited access to traditional foods and less financial capacity, resulting in a change of diet [38]. International students were also very vulnerable when the COVID-19 pandemic struck, as Australian government made no financial provisions for international students that left many students without any means of support [39]. In this context, international students have been reported by charitable agencies to be among the top newly food insecure groups in Australia throughout the pandemic [40]. As a high proportion of international students live in on-campus accommodation, with those on-campus having higher rates of food insecurity, the University has both the opportunity and the duty of care to support international student food security.

In our study, students who did not identify as either binary gender were at higher risk of food insecurity. We are the first to highlight this in the Australian context, but the results are similar to a study by the Wisconsin HOPE Lab, who reported that food insecurity among students with a non-binary gender was at 46%, which was significantly higher when compared to male-identifying students (28%) [41]. Another study in the US identified that the prevalence of food insecurity was 41–50% in transgender and non-binary students, which was substantially higher than male-identifying students (14.2–19.7%) and female identifying students (14.2–18.7%). Non-binary students have been shown to experience higher rates of poverty, joblessness, homelessness and discrimination which could be related to the increased risk of food insecurity observed in our study [42]. The unique needs of transgender and non-binary students should be explored and considered when planning campus interventions, since they may be particularly vulnerable to food insecurity and have less access to support resources. One study has suggested that universities could support safe, affirming resources for food involving a collaboration with transgender and non-binary student community organizations to provide “pop-up” food pantries in places that are easily accessible to people in need [43].

Our study was the first to show that staff on casual contracts, and newly recruited staff members were at higher risk of food insecurity. In Australia more broadly, “contractual vulnerability” was a major factor for those who lost jobs at universities throughout the pandemic, with a 10 per cent decline in the number of staff on fixed term contracts compared to 5 per cent for those with permanent contracts [44,45]. Indeed, nearly 60% of UTAS staff are engaged via casual employment [46], with a high proportion of casualisation evident across other Australian universities [47]. This factor is likely to be impacting long term food security in our sample of staff. Interestingly, our study also showed that academic staff employed on casual contracts experienced a similar occurrence of food insecurity (41%) compared with the student population (42%). This finding could be explained by the sporadic nature of casual employment for academic teaching staff, who sometimes have contracts that begin and end during the teaching semester, and whose hours of employment vary with enrolments. This means there could be several months of the year that these staff might not be employed or receive variable renumeration if they are otherwise employed. However, we were unable to explore this fully in the current study and this should be a consideration for future research. Overall, professional staff were at significantly higher risk of food insecurity compared to academic staff, which may relate to the relatively lower salaries held by this group (professional staff earning approximately 65–70% less than tenured academic staff at UTAS). Lastly, increasing length of employment was protective against food insecurity, independently to age. While employment status is a consistent predictor of food insecurity in the literature [48], our study is the first to demonstrate that length of employment is an important predictor of food insecurity in university staff. This has strong implications for advocacy efforts towards stable employment for members of university communities especially after the COVID-19 pandemic. At UTAS, in the two years since 2019 to the point of writing, about 1600 or more people lost employment despite the 2021 annual report documenting 170 million of dollars of revenue for the same financial year [46].

### 4.1. Potential Solutions to Food Insecurity in University Communities

The high occurrence of food insecurity in our study raises a major concern about campus sustainability efforts and prompts us to ask how universities can deliberately address the issue of food insecurity in their communities, and understand to what extent universities are addressing this issue. A whole systems setting approach to addressing food insecurity within universities is required through applying a food systems perspective.

Approaches to addressing food insecurity for staff and students are numerous and can be conceptualised across layers of the socio-ecological model adapted to the university setting. This public health model categorizes social issues into intrapersonal, interpersonal, organisational, and policy factors in order to promote health and social sustainability [49]. The model, overlayed with the six dimensions of food security [50], particularly highlights the importance of the pillar of agency, which puts students and staff at the front and centre of decision making. Actions across all levels of the model are required that focus on real, long-term solutions and move away from models of ‘relief’. Universities must continue to develop healthy, sustainable, and equitable food systems on campus, with benefits for both staff and students experiencing food insecurity.

### 4.2. Intrapersonal and Interpersonal Level Actions

Universities prioritise retention and completion rates for students, and high academic performance [51,52], however food insecurity compromises these outcomes. At an individual level, students and staff may be experiencing food insecurity due to issues with skills and confidence with acquiring and preparing enough safe and nutritious food, which is related to the ‘utilisation’ domain of food security [53]. This may be especially important for younger students who may be transitioning into early adulthood and experiencing individual level barriers in food procurement when transitioning out of the home environment [54]. Possible solutions at this level could involve universities establishing a peer-to-peer learning educational program to address these challenges, in addition to challenging attitudes and beliefs around food security. As international students were at higher risk of food insecurity in our study, incorporating understanding of cultural customs, and familiarisation with available food in the host setting, should also be considered. Importantly, promoting the agency of students and staff is key, where members of the university community experiencing food insecurity are able to actively participate in any decision-making about food provisioning on campus. Hearing the voices of food insecure staff and students will allow those with lived experience to shape their own relationships with the university food systems and begin to address the power imbalances that exist within university food systems [50].

### 4.3. University Organisational Level Actions

As university students and staff spend a considerable amount of time on campus, with students on -campus found to be at higher risk of food insecurity, the campus food environment is an important physical enabler or constraint for access to affordable and healthy food. Universities are well placed to provide strong leadership in promoting and supporting sustainable food systems through holistic institutional policies and governance mechanisms [55]. Policy and practice should focus on increasing stable and affordable access to adequate quantities and quality of food and addressing the structural causes of food insecurity. Change requires leaders with a deep understanding of the very real and critical social, environmental, and economic challenges associated with creating a sustainable food system on campus.

A key strategy would be for the tertiary sector to work towards establishing an overarching food policy and food strategy that incorporates the six dimensions of food security (availability, access, utilization, and stability, plus agency and sustainability) [50]. The policy could include on-campus dining and food service provisioning solutions to ensure the availability and equitable access (both physical and financial) to food on campus. Food pantries have been a commonly adopted model across universities internationally to deliver emergency food relief to those experiencing food insecurity [56]. At UTAS, there are food pantries on some campuses in addition to a program which provides bags of fresh fruit and vegetables to students at reduced rates. While effective to an extent, these initiatives may carry stigma and, when run by student organisations, they can experience challenges in maintaining student leadership and regular donations. Reliance on ‘food relief’ models of simply providing food to vulnerable students and staff is a short-term solution that is unable to address the root causes of food insecurity. Transitioning from this food relief model towards food social enterprises that are run on campus by those on campus could contribute to creating a food culture that increases the availability of more affordable food that meets the needs of those on-campus. Systemic solutions that build capacity, foster the campus community, and support a transition from a food relief to food resilience model are likely to be more self-sustaining in the longer-term. Creating an environment where students and staff can learn about food, from seed to plate, and gain hands-on learning opportunities through projects that improve the campus food system may be critical to this. Supporting circular food systems or a circular economy such as hosting markets on campus could contribute to providing a link between farmers and the university community for a more resilient food system.

Universities should also implement regular monitoring of food insecurity and conduct evaluations of campus food environments through standardised tools (such as the uni-food tool [57]) that can allow for benchmarking and comparison of the healthiness, sustainability, and equity of the food environment. Such benchmarks would provide an understanding of food availability and access related to food security. Food from on-campus providers should not compromise the environmental, health, economic and social wellbeing of present and future generations. Supporting all options for furthering access to healthy and sustainable food is important, which could include community gardens, urban farms, farmers markets, community supported agriculture, healthy food retailers, and new innovative means. Participation by students and staff in decision making about food provisioning on campus is critical. Incorporating an exploration of university staff and student experiences with the campus food environment is complimentary to this process, and could involve co-designing solutions for the university.

Lastly, as our study showed that casual staff and professional staff are vulnerable to food insecurity, initiatives that support and improve job security for university employees is a core strategy that could ultimately reduce food insecurity for university staff. There have been substantial profits (up to $AUD1 billion at one institution) recorded by Australian universities in 2022 [58], despite large numbers of staff layoffs during the pandemic that predominantly affected casual employees. Without action on this front, universities may be abrogating their ‘duty of care’ towards their staff.

### 4.4. Public Policy Level Actions

Policies and legislation at a local or national level that regulate or support food provision could support food security for vulnerable groups. Through supporting research on social determinants, universities could advocate for stronger social safety nets for vulnerable communities including university staff and students which would support access to food. Further, equivalent programs to the US Supplemental Nutrition Assistance Program (SNAP) a food assistance program that supports students (among other groups) do not exist in Australia [59], but support for such programs could be provided by universities.

### 4.5. Strengths and Limitations

The strengths of our study include using a validated tool to measure food insecurity that can measure marginal, low and very low food security. The inclusion of marginal food security in our estimates of food insecurity aligns with international recommendations and ensures our food security estimates best reflect the definition of food security. However, this should be noted when comparing with other studies which measure food insecurity using the HFSSM that may code marginal food security as food secure. While novel, our results must be considered within the limitations of the study. Our study was cross-sectional and therefore inferences are limited by the design of the study. We were unable to determine how the demographic characteristics of our respondents compared to that of the whole UTAS student and staff population. This means that despite our relatively large sample size and university-wide recruitment methods, it is possible our sample is not representative of the wider UTAS community and so should not be generalised beyond our sample. Some staff were not invited to participate including contract service staff, such as contract cleaners, security and staff from externally managed food services. Some casual staff members at UTAS are also students, which does not allow a clear delineation between the staff and student survey for this group. Adjunct staff in our study could comprise adjunct clinician researchers, tenured/contract staff aligned with another university, or unpaid staff with precarious employment who are distinct in terms of food security risk. Future research may consider examining total income of staff to examine the difference in food insecurity between adjuncts more clearly. Lastly, while our study gives some insight into the impact of food insecurity on the quality and quantity of food consumed, future research could focus on the impact of food insecurity on health and education/employment outcomes for students and staff.

## 5. Conclusions

In conclusion, this study offers new information regarding the severity of food insecurity in university students, professional staff, and academic staff at a regional Australian university. Our findings suggest many members of the university community experience food insecurity, and a high proportion of food insecure staff and students experience very low and very low food security. Our results serve as an impetus for Australian universities to continue to examine the experience of food insecurity across the entire university community. Through our identification of groups of students and staff at risk of food insecurity, universities have the opportunity to intervene, potentially increasing retention and health outcomes for vulnerable staff and students. Universities are well-placed to provide strong organisational leadership in promoting and supporting sustainable food systems through holistic institutional policies and governance mechanisms that address the intrapersonal, interpersonal, university organisational and public policy factors that influence food security. Our findings could inform efforts towards reducing food insecurity at university campuses through a holistic and integrated approach, advocating for food systems that support healthy, sustainable, and equitable food procurement and provision that address the needs of both university students and staff.

## Figures and Tables

**Figure 1 nutrients-14-03956-f001:**
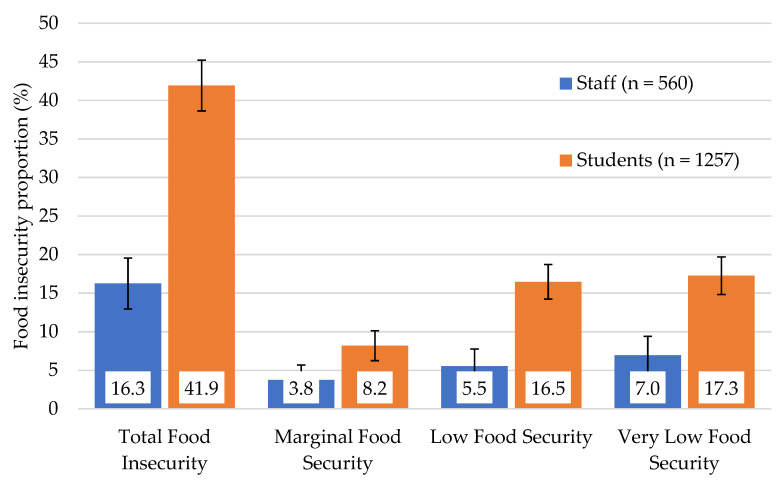
Student (*n* = 1257) and Staff (*n* = 560) Food Security Status according to the six-item HFSSM (error bars represent 95% Confidence Intervals).

**Figure 2 nutrients-14-03956-f002:**
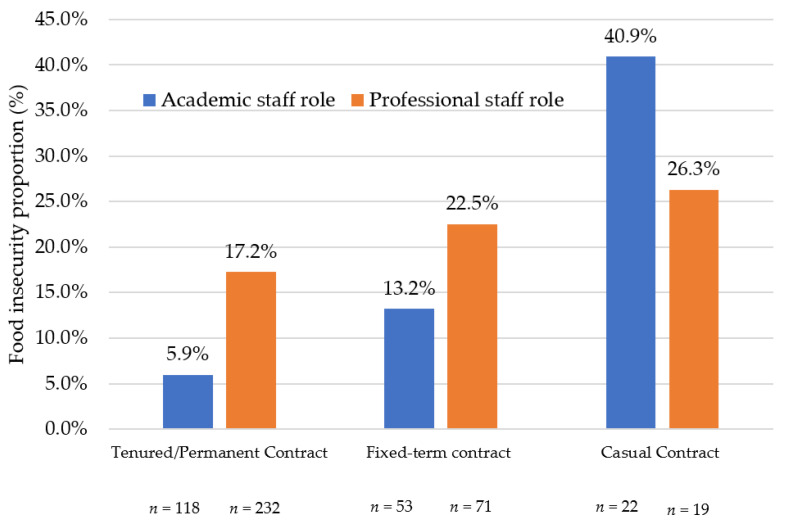
Percentage of academic and professional staff experiencing food insecurity according to employment contract type.

**Table 1 nutrients-14-03956-t001:** Food Security Status of the sample of university students according to demographic and education characteristics.

Characteristic			Food Security Status			
		Total *n* (%)	High Food Security *n* (%)	Marginal Food Security *n* (%)	Low Food Security *n* (%)	Very Low Food Security *n* (%)
Age in years (*n* = 1249)	18–24	531 (42.5)	276 (52.0)	102 (19.2)	54 (10.2)	99 (18.6)
	25–34	328 (26.3)	174 (53.0)	52 (15.9)	27 (8.2)	75 (22.9)
	35–54	283 (22.7)	192 (67.8)	44 (15.5)	16 (5.7)	31 (11.0)
	55+	107 (8.6)	83 (77.6)	8 (7.5)	5 (4.7)	11 (10.3)
Gender (*n* = 1257)	Man or Male	347 (27.6)	195 (56.2)	62 (17.9)	29 (8.4)	61 (17.6)
	Woman or Female	868 (69.1)	522 (60.1)	137 (15.8)	67 (7.7)	142 (16.4)
	Non-binary, self-identify or prefer not to disclose	42 (3.3)	13 (31.0)	8 (19.0)	7 (16.7)	14 (33.3)
Years of enrolment (*n* = 1257)	First year	526 (41.8)	285 (54.2)	100 (19.0)	43 (8.2)	98 (18.6)
	Second Year	313 (24.9)	185 (59.1)	52 (16.6)	27 (8.6)	49 (15.7)
	Third year or longer	418 (33.3)	260 (62.2)	55 (13.2)	33 (7.9)	70 (16.7)
Degree of enrolment (*n* = 1257)	Pre-degree or short course	369 (29.4)	224 (60.7)	51 (13.8)	28 (7.6)	66 (17.9)
	Undergraduate (including honours)	568 (45.2)	315 (55.5)	105 (18.5)	56 (9.9)	92 (16.2)
	Postgraduate	320 (25.5)	191 (59.7)	51 (15.9)	19 (5.9)	59 (18.4)
Mode of study (*n* = 1257)	On campus	713 (56.7)	369 (51.8)	134 (18.8)	68 (9.5)	142 (19.9)
	Distance	544 (43.3)	361 (66.4)	73 (13.4)	35 (6.4)	75 (13.8)
Location of usual campus for on-campus students only (*n* = 713)	South	487 (68.3)	254 (52.2)	98 (20.1)	37 (7.6)	98 (20.1)
	North	178 (25.0)	91 (51.1)	28 (15.7)	21 (11.8)	38 (21.3)
	Northwest	29 (4.1)	15 (51.7)	5 (17.2)	4 (13.8)	5 (17.2)
	Sydney	19 (2.7)	9 (47.4)	3 (15.8)	6 (31.6)	1 (5.3)
Enrolment status (*n* = 1257)	Domestic	1064 (84.6)	654 (61.5)	167 (15.7)	79 (7.4)	164 (15.4)
	International	193 (15.4)	76 (39.4)	40 (20.7)	24 (12.4)	53 (27.5)
Total sample			730 (58.1)	103 (8.2)	207 (16.5)	217 (17.3)

**Table 2 nutrients-14-03956-t002:** Univariate and multivariate logistic regression results of food security with demographic characteristics in a sample of Australian university students.

Characteristic		Univariate Regression				Multivariate Regression			
		Odds Ratio	SE	95% CI	*p*-Value	Adjusted Odds Ratio	SE	95% CI	*p*-Value
Age in years (*n* = 1249)	18–24	3.20	0.25	[1.97, 5.19]	<0.001	2.22	0.27	[1.32, 3.74]	<0.001
	25–34	3.08	0.26	[1.86, 5.09]	<0.001	2.15	0.27	[1.27,3.62]	<0.001
	35–54	1.64	0.26	[0.98, 2.75]	0.06	1.43	0.27	[0.85, 2.41]	0.18
	55+	Reference category			Reference category				
Gender (*n* = 1257)	Man or Male	Reference category			Reference category				
	Woman or Female	0.85	0.13	[0.66, 1.10]	0.21	0.99	0.14	[0.76, 1.29]	0.95
	Non-binary, self-identify or prefer not to disclose	2.86	0.35	[1.44, 5.69]	<0.001	3.46	0.38	[1.64, 7.31]	<0.001
Years of enrolment (*n* = 1257)	First year	1.39	0.13	[1.07, 1.81]	0.01	1.43	0.14	[1.08, 1.88]	0.01
	Second Year	1.15	0.15	[0.85, 1.55]	0.38	1.17	0.16	[0.85, 1.60]	0.33
	Third year or longer	Reference category			Reference category				
Degree of enrolment (*n* = 1257)	Pre-degree or short course	Reference category				-	-	-	-
	Undergraduate (including honours)	1.24	0.14	[0.95, 1.62]	0.11	-	-	-	-
	Postgraduate	1.05	0.16	[0.77, 1.42]	0.76	-	-	-	-
Mode of study (*n* = 1257)	On campus	1.83	0.12	[1.46, 2.31]	<0.001	1.33	0.15	[1.00, 1.77]	0.05
	Distance	Reference category			-	Reference category			
Location of usual campus for on-campus students only (*n* = 713)	South	Reference category				-	-	-	-
	North	1.04	0.18	[0.74, 1.47]	0.81	-	-	-	-
	Northwest	1.02	0.38	[0.48, 2.15]	0.96	-	-	-	-
	Sydney	1.21	0.47	[0.48, 3.03]	0.68	-	-	-	-
Enrolment status (*n* = 1257)	Domestic	Reference category				-	-	-	-
	International	2.45	0.16	[1.79, 3.36]	<0.001	1.94	0.18	[1.36, 2.75]	<0.001

**Table 3 nutrients-14-03956-t003:** Food Security Status of the sample of university staff according to demographic and employment characteristics.

Characteristic			Food Security Status			
		Total	High Food Security *n* (%)	Marginal Food Security *n* (%)	Low Food Security *n* (%)	Very Low Food Security *n* (%)
Age in years (*n* = 540)	18–34	90 (16.7)	66 (73.3)	5 (5.6)	11 (12.2)	8 (8.9)
	35–44	151 (28.0)	118 (78.1)	9 (6.0)	9 (6.0)	15 (9.9)
	45–54	153 (28.3)	138 (90.2)	2 (1.3)	6 (3.9)	7 (4.6)
	55–64	146 (27.0)	131 (89.7)	4 (2.7)	4 (2.7)	7 (4.8)
Gender (*n* = 556)	Man or Male	221 (39.7)	185 (83.7)	8 (3.6)	13 (5.9)	15 (6.8)
	Woman or Female	310 (55.8)	262 (84.5)	12 (3.9)	16 (5.2)	20 (6.5)
	Non-binary, self-identify or prefer not to disclose	25 (4.5)	19 (76.0)	1 (4.0)	2 (8.0)	3 (12.0)
Type of role (*n* = 559)	Academic	207 (37.0)	184 (88.9)	8 (3.9)	5 (2.4)	10 (4.8)
	Professional	328 (58.7)	266 (81.1)	12 (3.7)	23 (7.0)	27 (8.2)
	Academic and professional	24 (4.3)	18 (75.0)	1 (4.2)	3 (12.5)	2 (8.3)
Employment contract (*n* = 556)	Tenured or Permanent	357 (64.2)	308 (86.3)	12 (3.4)	16 (4.5)	21 (5.9)
	Fixed Term contract	130 (23.4)	106 (81.5)	5 (3.8)	9 (6.9)	10 (7.7)
	Casual	49 (8.8)	32 (65.3)	4 (8.2)	6 (12.2)	7 (14.3)
	Adjunct/Honorary	20 (3.6)	19 (95.0)	0 (0.0)	0 (0.0)	1 (5.0)
Location of usual campus (*n* = 559)	South	392 (70.1)	329 (83.9)	13 (3.3)	22 (5.6)	28 (7.1)
	North	131 (23.4)	112 (85.5)	4 (3.1)	8 (6.1)	7 (5.3)
	Northwest	17 (3.0)	14 (82.4)	1 (5.9)	0 (0.0)	2 (11.8)
	Sydney	9 (1.6)	7 (77.8)	1 (11.1)	1 (11.1)	0 (0.0)
	Not associated with one region/Distance	10 (1.8)	6 (60)	2 (20)	0	2 (20)
Length of employment (*n* = 559)	Less than 1 year	61 (10.9)	42 (68.9)	4 (6.6)	7 (11.5)	8 (13.1)
	1–3 years	79 (14.1)	59 (74.7)	5 (6.3)	7 (8.9)	8 (10.1)
	4–6 years	115 (20.6)	95 (82.6)	3 (2.6)	6 (5.2)	11 (9.6)
	7–9 years	93 (16.6)	79 (84.9)	6 (6.5)	4 (4.3)	4 (4.3)
	10+ years	211 (37.7)	193 (91.5)	3 (1.4)	7 (3.3)	8 (3.8)
Total sample			469 (83.8)	21 (3.8)	31 (3.7)	39 (4.7)

**Table 4 nutrients-14-03956-t004:** Univariate and multivariate logistic regression results of food security with demographic characteristics in a sample of Australian university staff.

Characteristic		Univariate Logistic Regression				Multivariate Logistic Regression			
		Odds Ratio	SE	95% CI	*p*-Value	Adjusted Odds Ratio	SE	95% CI	*p*-Value
Age in years (*n* = 540)	18–34	3.18	0.36	[1.56, 6.46]	0.00	-	-	-	-
	35–44	2.44	0.34	[1.26, 4.72]	0.01	-	-	-	-
	45–54	0.95	0.39	[0.45, 2.02]	0.89	-	-	-	-
	55 years+	Reference category				-	-	-	-
Gender (*n* = 556)	Man or Male	Reference category				-	-	-	-
	Woman or Female	0.94	0.24	[0.59, 1.51]	0.80	-	-	-	-
	Non-binary, self-identify or prefer not to disclose	1.62	0.50	[0.61, 4.35]	0.34	-	-	-	-
Type of role (*n* = 559)	Academic	Reference category				-	-	-	-
	Professional	1.87	0.26	[1.12, 3.12]	0.02	1.82	0.28	[1.06, 3.12]	0.03
	Academic and professional	2.67	0.52	[0.96, 7.40]	0.06	2.34	0.56	[0.78, 6.99]	0.13
Employment contract (*n* = 556)	Tenured or Permanent	Reference category				Reference category			
	Fixed Term contract	1.42	0.27	[0.83, 2.43]	0.20	1.19	0.29	[0.67, 2.10]	0.55
	Casual	3.34	0.34	[1.72, 6.47]	<0.001	2.45	0.38	[1.17, 5.11]	0.02
	Adjunct/Honorary	0.33	1.04	[0.04, 2.53]	0.29	0.50	1.07	[0.06, 4.03]	0.51
Location of usual campus (*n* = 559)	South	Reference category				-	-	-	-
	North	0.89	0.28	[0.51, 1.55]	0.67	-	-	-	-
	Northwest	1.12	0.65	[0.31, 4.01]	0.86	-	-	-	-
	Sydney	1.49	0.81	[0.30, 7.35]	0.62	-	-	-	-
	Distance	3.48	0.66	[0.96, 12.69]	0.06	-	-	-	-
Length of employment (*n* = 559)	Less than 1 year	4.85	0.37	[2.35, 10.0]	<0.001	3.63	0.40	[1.67, 7.87]	<0.001
	1–3 years	3.64	0.36	[1.80, 7.32]	<0.001	2.83	0.37	[1.36, 5.88]	0.01
	4–6 years	2.26	0.35	[1.14, 4.47]	0.02	1.92	0.36	[0.95, 3.85]	0.07
	7–9 years	1.90	0.38	[0.90, 4.01]	0.09	1.67	0.39	[0.79, 3.56]	0.18
	10+ years	Reference category				Reference category			

## Data Availability

Data supporting this manuscript is not publicly available but can be made available upon reasonable written request to the corresponding author (KK).

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
