# Peer review of "Severity of Food Insecurity among Australian University Students, Professional and Academic Staff"

_nutrients, 2022, doi:10.3390/nu14193956_

Round 1

Reviewer 2 Report

In this study, a cross-sectional online survey in March 2022 aimed to characterize the severity of food insecurity in students, professional and academic staff at the University of Tasmania (UTAS). The topic is interesting. However, there are some problems in the study. The experiment design is too simple. Some important information is missing. For example, it is important to provide the number, age composition and gender of staff, and the number and gender of students in this survey.

In addition, it is important to provide the types of food and the types and source of unsafe food which are vital to food safety survey because it is meaningless to investigate food safety without knowing the specific types of unsafe food. Moreover, what is the basis for scores as having high (0), marginal (1), low (2 - 4) or very low food security (5 - 6)?

Round 2

Reviewer 2 Report

The authors are unwilling or unable to address my concerns sufficiently to make this manuscript suitable for publication.